# Advancing Exoskeleton Development: Validation of a Robotic Surrogate to Measure Tibial Strain

**DOI:** 10.3390/bioengineering11050490

**Published:** 2024-05-15

**Authors:** Robert L. McGrath, Ciera A. Price, William Brett Johnson, Walter Lee Childers

**Affiliations:** 1The Henry M. Jackson Foundation for the Advancement of Military Medicine, Inc., 6720A Rockledge Dr, Bethesda, MD 20817, USA; 2Center for the Intrepid, Brooke Army Medical Center, 3551 Roger Brooke Drive, Fort Sam Houston, TX 78234, USA; 3DoD/VA Extremity Trauma and Amputation Center of Excellence, Defense Health Agency, Falls Church, VA 22042, USA

**Keywords:** exoskeleton, bone stress injury, surrogate, ankle–foot orthosis, tibial strain, stiffness

## Abstract

Bone stress injuries are prevalent among athletes and military recruits and can significantly compromise training schedules. The development of an ankle–foot orthosis to reduce tibial load and enable a faster return to activity will require new device testing methodologies capable of capturing the contribution of muscular force on tibial strain. Thus, an actuated robotic surrogate leg was developed to explore how tibial strain changes with different ankle–foot orthosis conditions. The purpose of this work was to assess the reliability, scalability, and behavior of the surrogate. A dual actuation system consisting of a Bowden cable and a vertical load applied to the femur via a material testing system, replicated the action-reaction of the Achilles-soleus complex. Maximum and minimum principal strain, maximum shear strain, and axial strain were measured by instrumented strain gauges at five locations on the tibia. Strains were highly repeatable across tests but did not consistently match in vivo data when scaled. However, the stiffness of the ankle–foot orthosis strut did not systematically affect tibial load, which is consistent with in vivo findings. Future work will involve improving the scalability of the results to match in vivo data and using the surrogate to inform exoskeletal designs for bone stress injuries.

## 1. Introduction

Bone stress injuries (BSIs) can occur in any physically active person but are most prevalent among athletes and military recruits [1]. If left unresolved, a BSI will eventually develop into a fatigue stress fracture through the continued application of repetitive stress on bone. Stress fractures are most common in the tibia and midfoot but have also been observed in the spine, pelvis, and upper extremities depending on the stressful activity [1,2]. Incidence rates vary according to age, gender, ethnicity, fitness level, body mass index, type of physical activity, bone geometry, and biomechanics [3,4,5,6,7,8,9]. Runners have an incidence rate of 15%, with BSIs accounting for 70% of all their injuries [2]. Military recruits also have high incidence rates due to the rapid onset of repetitive, high-intensity training, particularly running and marching [1,3,10]. An analysis of BSI rates among all active components of the U.S. Armed Forces found that recruits developed BSIs at a rate over eighteen times that of non-recruits. Furthermore, the tibia/fibula was the most affected anatomical site, with BSI incidence rates of 13.21% and 29.8% among male and female recruits, respectively [1,3]. Conventional management of BSIs requires a prolonged period of recovery and activity modification, lasting anywhere from 4–18 weeks [1,11,12], which equates to an average of 85 lost duty days for military personnel [13]. BSIs are responsible for more lost duty days and delayed training than any other training-related injury [3] and pose a significant threat to military readiness [11]. Use of orthotic devices, such as a pneumatic walking boot, may reduce recovery time to an extent [1,2,14]. However, the current standard of care remains inadequate to return Service members to full activity within the time constraints of a military training schedule.

Advancements in the orthotic management of BSIs will require innovation not only in design, but also in device testing and validation. The Center for the Intrepid (CFI) [15] is a multi-disciplinary rehabilitation and research facility working to develop a novel ankle–foot orthosis (AFO) to reduce tibial strain. A physical model for testing and validation will be required for the development of this type of AFO. Typically, cadaveric modeling is the standard testing methodology for orthoses and other medical devices prior to human subject testing [16]. However, a cadaveric model cannot capture the significant contribution of plantarflexor muscle forces on tibial force (Figure 1a) [17,18] and therefore tibial stress and the associated strain. Human surrogates have been developed extensively to test and iterate personal protective equipment, but these also fail to meet our needs for evaluating tibial strain [19]. Specifically, lower-extremity surrogates have been developed for the purposes of evaluating lower leg injury from floor impact due to landmine explosions [20,21]. Moreover, a lower-extremity surrogate composed of 3D-printed bones in ballistics gel has effectively emulated kinematics for the evaluation of active knee exoskeletons, but ballistics gel replicates the density of human soft tissue rather than the stiffness [22].

Surrogates have been devised to evaluate AFOs [23] but, to our knowledge, surrogates for the assessment of tibial strain under replicated anatomical loads have yet to be developed and implemented. We developed a robotic lower-extremity surrogate with the capability of applying anatomical plantarflexor muscle action/reaction forces and a vertical external force caused by the mass of the body (Figure 1b). The purpose of this work was to (1) assess the reliability of the robotic surrogate in load application and resulting tibial strain, (2) assess the scalability of tibial strains given the application of a percentage of anatomical loads, and (3) compare the behavior of the surrogate leg in AFOs of varying strut stiffnesses to relevant in vivo data.

## 2. Methods and Materials

### 2.1. The Robotic Surrogate Design

We devised a novel robotic surrogate consisting of a musculoskeletal apparatus (Figure 2) encased in Dragon Skin 10NV silicone rubber (Smooth-on, Macungie, PA, USA). The silicone was thinned at 40% to better replicate the stiffness of human soft tissue, and poured into a mold the size of a male leg roughly within the 50th percentile of male U.S. Army Service members according to knee height and calf circumference [24]. A model tibia, fused foot bones, and distal femur section were printed from Rigid 10K resin (Formlabs Inc., Somerville, MA, USA) with a Young’s modulus (10 GPa) comparable to that of trabecular bone (10.4 GPa) [25,26] and cortical bone (10.1–18.2 GPa) [25]. The comparable Young’s modulus is a critical material property for the robotic surrogate design as the quantified tibial surface strain is the primary outcome measure. Five 45-degree strain gauge rosettes (VPG, Inc., Raleigh, NC, USA) were adhered with M-Bond 200 (VPG, Inc., Raleigh, NC, USA) to the surface of the tibia at five key locations. BSIs have been observed to occur primarily in the middle and distal thirds of the tibia [27,28] and at the posteromedial cortex [29]. In vivo assessments of strain were acquired on the medial aspect of the middle and distal third of the tibia [30]. As such, we chose the medial distal (MedDist), medial middle (MedMid), lateral middle (LatMid), posterior middle (PostMid), and medial proximal (MedProx) locations along the tibia.

A three-piece rotary ankle joint was machined from aluminum and fitted with a 15T Accu-coder low-profile rotary encoder (Encoder Products Company, Sagle, ID, USA) to track the ankle angle. A wedge of ethylene-vinyl acetate (EVA) foam was used to achieve a neutral coronal alignment between the femur and tibia. Epoxy (Fabtech, Everett, WA, USA) was applied to the tibial plateau to rigidly fix the distal femur to the proximal tibia. The bond was reinforced with a circumferential wrap of Delta-Lite Plus Fiberglass Cast Tape (BSN Medical, Charlotte, NC, USA), forming a fixed knee joint. The proximal section of the femur was fit with a steel rod for mounting in a material testing system (MTS) (Instron Corp., Norwood, MA, USA). The proximal end of the inner Bowden assembly V-12 Vectran single-braid cable (New England Ropes, Fall River, MA, USA) was attached to the posterior superior aspect of the calcaneus. The proximal end of the outer housing (Lexco Cable, Norridge, IL, USA) was attached to the posterior tibia, approximately one-third of the tibial length from the tibial plateau. These attachment points replicate the insertion points of the Achilles tendon and origin of the soleus muscle, respectively.

Outside of the robotic surrogate, a custom breakout tube and linear transducer (Futek Inc., Irvine, CA, USA) were installed in series with the Bowden assembly to allow force feedback. The distal end of the Bowden assembly was attached to the Caplex system (Humotech LLC, Pittsburgh, PA, USA) to allow for high-fidelity cable tension application. This Bowden assembly arrangement allowed for an action–reaction tension application for the replication of the Achilles-soleus complex. The fabrication manual is in Appendix A(Appendix A).

### 2.2. The Test Device

The Intrepid Dynamic Exoskeletal Orthosis (IDEO) (Figure 3) is a passive dynamic AFO for managing high-energy lower-extremity trauma that was previously developed at the CFI. Our clinic has over a decade of clinical experience fitting this device, which has demonstrated improved functional outcomes in agility, plantarflexor power, and speed relative to conventional orthotic management [31]. The deflection of the IDEO during single-limb support absorbs energy which is then returned as ankle power once the limb is unloaded in pre-swing [32]. If the energy storage and return capabilities of the IDEO can reduce plantarflexor muscle activity [33], the resulting decrease in plantarflexor muscle force could reduce tibial strain [17]. Therefore, the IDEO was used to evaluate the surrogate because it has been tested extensively in humans and is thought to alter tibial strain, providing an assessment of surrogate responsiveness.

In clinical practice, the IDEO is commonly prescribed to address impaired plantarflexion and propulsive ability, joint instability, and painful weight-bearing [31]. These functional deficits may stem from a number of pathologies, with ankle injuries, tibia fractures, and nerve injuries being the most common diagnoses [34]. Each IDEO is custom-fabricated to address the user’s specific needs; factors such as ankle stability and pain-free range of motion dictate trimlines and other design considerations [35]. All IDEOs are fabricated through a multi-stage carbon lamination process and characterized by a proximal cuff, posterior strut, and distal footplate.

The cuff offloads the distal limb and consists of a hinged anterior panel that resembles the proximal aspect of a patella-tendon-bearing transtibial socket [34]. There are two types of posterior struts currently being used with the IDEO: the interchangeable Posterior Dynamic Element (PDE) spring (Fabtech, Everett, WA, USA) and the fixed Clever Bone Rod (Ossur Americas, Irvine, CA, USA) [32]. The PDE struts were used in this study and are available in three lengths and seven stiffness categories ranging from 1 (least stiff) to 7 (most stiff) [36]. The stiffness category is determined by the clinician according to the user’s weight and activity goals but does not require precise tuning for optimal walking [33,37] or running performance [35]. The distal footplate is characterized by supra-malleolar trim lines, a gradual rollover shape, and a plantarflexed foot position carefully selected to maintain the ankle–foot complex in a pain-free position [32,35]. Proper sagittal alignment requires the incorporation of a compressible heel wedge which provides shock absorption and controlled forward progression of the tibia [34].

A modified IDEO was custom fabricated for the robotic surrogate and utilized for testing. The hinged proximal cuff did not facilitate easy donning as human wearers are required to plantarflex the ankle and this surrogate could not achieve such a plantarflexed position for donning. A proximal cuff of comparable shape with a removable anterior panel was fabricated for the test device. Moreover, 300 mm long PDE springs of varying stiffnesses were used without modification. The footplate had a greatly reduced forefoot rocker to make it more compatible with a combat boot and it was not paired with a heel wedge due to a focus on late stance, rendering the heel wedge redundant. The overall fabrication process and materials remained unchanged.

### 2.3. Mechanical Testing Setup

The mechanical testing setup consisted of the robotic surrogate mounted in the Instron MTS, which applied a cyclical vertical force (VF) profile, while the Caplex system applied a synchronized cyclical Achilles tendon force (ATF) profile (Figure 4). The tibial strain signals were acquired at 1000 Hz for offline processing. The MTS utilized proportional–integral–derivative force-feedback control tuned to specimen stiffness to achieve the desired VF profile. The VF was derived from a simplified quasistatic model of the leg based on an ankle moment extracted from the literature [38] and scaled to a specified percentage of anatomical moment. A quasistatic model converted the ankle moment to an equivalent VF given the lever arm distance (18 cm) between the medial malleolus and approximately the fifth metatarsal head—the point of contact of the foot with a sine plate. A digital trigger was transmitted from the Caplex system to the strain acquisition system System 8000 (VPG, Inc., Raleigh, NC, USA) and MTS to synchronize all three systems. For each test condition, the robotic surrogate was fit with a U.S. men’s size 12 standard issue boot (T8 Bifida, Garmont International North America, Inc, Portsmouth, NH, USA) with or without an IDEO with a specified strut stiffness. Each test condition was run for 6 separate trials, each consisting of 20 gait cycles at a frequency of 1 Hz.

Repeatability Evaluation: Testing was performed to determine if doffing and donning the IDEO and boot on the robotic surrogate affect tibial strain. A set of trials was acquired under 50% anatomical load in which the IDEO with a category 4 PDE strut and standard issue boot were doffed and donned between each of the six trials. For comparison, a separate set of six trials was collected at 50% anatomical load without doffing and donning the IDEO and boot in between trials.

Strain Scaling Evaluation: Testing was performed to determine if tibial strain scales linearly with percent anatomical load and if the acquired strain is comparable to in vivo assessments of tibial strain. To achieve this, a set of trials in the standard issue boot-only condition under 25%, 37.5%, and 50% anatomical loads were conducted.

Stiffness Evaluation: Testing was performed to determine the effect of PDE strut stiffness on tibial strain. Four separate test conditions under 50% anatomical load were collected. This included the robotic surrogate fit with a standard issue boot and an IDEO with one of three categorized PDE struts (1, 4, or 7) that spanned the full range of available stiffnesses. A set of boot-only trials was conducted to serve as a control condition.

### 2.4. Data Analysis

Following fabrication and experimental testing (Figure 5), multiple analyses were run to evaluate the surrogate’s performance. The four time-series signals of maximum principal strain (Max PS), minimum principal strain (Min PS), maximum shear strain (MSS), and axial strain (AS) were exported offline to MATLAB (MathWorks, Inc., Natick, MA, USA) and filtered with a 15 Hz cutoff fourth-order Lowpass Butterworth filter. For Max PS, Min PS, and MSS, ten extrema strain values were extracted (cycles 10–19) from these time-series signals for each strain sensor location (MedDist, MedMid LatMid, PostMid, and MedProx) for each of the six trials, for all test conditions.

Repeatability Evaluation: To evaluate the effect of doffing a device between trials, JMP Pro 16 (SAS Institute, Cary, NC, USA) was used to calculate the intraclass correlation coefficient (ICC) across the three outcome measures of Max PS, Min PS, and MSS at all five sensor locations.

Strain Scaling Evaluation: For AS, ten minima and ten maxima were extracted from (cycles 10–19) for each of the six trials, for each anatomical load condition (35%, 37.5%, and 50%) with the standard issue boot. The mean and standard deviation of AS minima and maxima were measured for all five sensor locations across all anatomical loads for all six trials. Lines of best fit were used to extrapolate the strains at 100% anatomical load. The VF and ATF accuracy were evaluated as the mean and SD of the measured force divided by the target force across all scaled loads and all trials.

Stiffness Evaluation: We utilized a generalized linear mixed-model analysis in JMP Pro 16 for Max PS, Min PS, and MSS. The random effect was the trial number (1–6) and the fixed effects were the test condition (Boot Only, PDE strut categories 1, 4, and 7) and strain sensor location (MedDist, MedMid, LatMid, PostMid, and MedProx). The continuous fixed effects of measured VF applied by the MTS were included in the model to account for variability in force application. All main effects, two-way effects, and the three-way effect were included in the model. We performed post hoc Tukey HSD pairwise comparisons to ascertain significant differences across test conditions, within the sensor location, with a significance threshold of α = 0.05.

## 3. Results

The system successfully applied synchronized scaled ATF and VF profiles (Figure 6a), resulting in repeatable cycles of strain time-series signals being acquired at each of the five sensor locations (Figure 6b).

Repeatability Evaluation: The ICC across the three outcome strain measures and five strain locations was 0.962, indicating high reliability in strain values across doff/don and no doff/don conditions (Figure 7).

Strain Scaling Evaluation: The Caplex system achieved a high force fidelity of 98.9 ± 2.1% across all conditions, and the MTS achieved a reasonable force fidelity of 97.6 ± 6.9% across all scaled loads and trials (Figure 8a,b). The MedDist and MedMid minimum and maximum strains scaled linearly with anatomical load percentage (Figure 8c). The projected axial strain extrema at 100% anatomical load in the boot-only conditions are shown (Table 1).

Stiffness Evaluation: The statistical models fitted to Max PS (Figure 9a), Min PS (Figure 9b), and MSS (Figure 9c) all featured very high adjusted R^2^ values (0.987, 0.989, and 0.988, respectively). The random effects of the trial were insignificant for all three statistical models. Except for the main effect of VF for Max PS, all main effects, two-way interactions, and three-way interactions for the test condition, sensor location, and VF were significant for all three models (*p* < 0.001). Significant pairwise comparisons (*p* < 0.05) within the sensor location and between boot only and each of the conditions of PDE strut categories 1, 4, and 7 are indicated with asterisks in Figure 9. In all strut conditions compared to boot only, Max PS decreased at the MedDist, MedMid, LatMid, and PostMid sensor locations. Max PS increased at the MedProx sensor location for PDE strut categories 4 and 7. In all strut conditions compared to boot only, Min PS decreased at the LatMid and PostMid sensor locations and increased at the MedDist, MedMid, and MedProx sensor locations. In all strut conditions compared to boot only, MSS decreased at the MedDist, LatMid, and PostMid sensor locations and increased at the MedMid and MedProx sensor locations.

## 4. Discussion and Conclusions

In this work, we designed a lower-extremity robotic surrogate capable of applying 50% scaled plantarflexor and vertical loads. We measured tibial strain at five key surface locations in a boot-only and IDEO condition with three different PDE strut stiffness categories. Firstly, our results demonstrate repeatable data across testing cycles when donning and doffing the device, as indicated by a high ICC value of 0.962. The large compressive strain at the PostMid sensor measured with our robotic surrogate indicates a posterior bending moment (Figure 6). This effect is also observed in previous in vitro work modeling tibial strain during exercise in microgravity [39] in which posterior and anterior sensors are under compression and tension, respectively. Discrepancies between the medial and lateral surface sensors of the two works could be due to differences in geometry and an effective anterior–posterior axis of actuation and/or exact sensor placement. This replication of a posterior bending moment with our robotic surrogate for use in BSI device development is important as early work determined that the area moment of inertia is a risk factor for tibial BSI [40]. This is due to the fact that a larger bending moment about the anterior–posterior axis would resist the natural bending moment that induces stress and therefore strain in the medial cortex [40].

Secondly, the MedDist and MedMid tibial sensor locations of this work were spatially comparable to the “distal” and “proximal” locations, respectively, measured in an in vivo study from three participants during treadmill walking at 5 km/h [30]. Specifically, we compare the peak axial strain values due to differences in walking cadence and our focus on the propulsive phase of gait. The participants’ peak axial tensile strain in the distal location ranged between 532.5 ± 19 µε and 865.9 ± 27.6 µε [30], while the MedDist location was extrapolated to 399.7 µε at 100% load. The participants’ peak axial tensile strain in the proximal location ranged between 90.0 ± 27.4 µε and 439.8 ± 39.4 µε [30], and in this work, the tensile strain at the MedMid location was extrapolated to 123.4 µε at 100% load. The participants’ peak axial compressive strain in the proximal location ranged between −135.4 ± 29.6 µε to −696.6 ± 22.9 µε [30] while in our work the MedMid location was extrapolated to −86.7 µε at 100% load. Importantly, the MedMid location under tensile strain, scaled linearly, is well within range of the in vivo data. However, the peak axial tension and peak axial compression at the MedDist location did not reach the values measured in vivo. It is apparent that our model does not reach large compressive axial loads at either of the two locations of interest, which may be related to the in vivo system having additional muscles we did not replicate (i.e., posterior tibialis), and this requires further exploration.

Thirdly, the large changes in strain from the boot to an IDEO of any stiffness are consistent with research conducted in humans using similar AFO designs [41]. For example, Corlett [41] found significantly lower peak forefoot forces in the Firm and Mod CDO conditions relative to the no brace and MLSO conditions. This is in parallel with our finding that the introduction of the IDEO resulted in a large reduction in strain, further supporting this robotic surrogate as a model for device development. In addition, we observed minimal changes in strain across the different PDE strut conditions which is consistent with existing literature describing the influence of AFO stiffness on walking and running mechanics. Russell Esposito et al. [37] previously demonstrated that a 40% range in strut stiffness had minimal impact on temporal–spatial parameters (e.g., stride length), peak kinematics (e.g., joint angles), or peak kinetics (e.g., joint powers) for IDEO users walking at a self-selected velocity. Parallel studies reported comparable findings during both walking [33] and running [35], suggesting that there is a range of clinically appropriate strut stiffnesses that will result in comparable biomechanical outcomes. Moreover, strut stiffness has been shown to have negligible effect on ankle plantarflexor electromyography (EMG). A systematic literature review on AFO stiffness and gait in diverse patient populations reported little evidence that stiffness affects peak gastrocnemius or soleus EMG activation [42]. In addition, Corlett found no significant differences in plantarflexor EMG activation in healthy subjects across four bracing conditions, including a no-brace condition and three AFOs of varying stiffness [43]. In our work, we did not vary muscle force to best replicate the lack of change in muscle activity presented in earlier work [42,43] and demonstrated tibial strain values consistent with outcomes from previous studies related to bone loading [33,35,37]. This reinforces that the parallel pathway for mechanical energy enabled by an external device is sufficient to reduce bone loading.

This robotic surrogate could therefore be used to elicit potential mechanisms that enable reduced bone loading before proceeding to human testing to confirm the findings. Furthermore, the robotic surrogate can be used to measure the effect of changes in AFO characteristics unrelated to strut stiffness, which may have a more significant influence on limb loading. Corlett [43] demonstrated significant changes in plantar forces with three different AFO designs. Specifically, the least-stiff AFO lacked a proximal cuff and full-length footplate, while the other two AFOs had identical cuffs and full-length footplates. There were numerous significant differences between the devices with different trimlines but only one significant difference between the two devices with identical trimlines and different stiffnesses. Thus, while the existing literature supports that strut stiffness may not impact walking biomechanics or EMG activity, it is reasonable that other AFO design characteristics (materials, alignment, trimlines, etc.) could influence these outcomes. As such, the robotic surrogate provides a testbed for quickly eliciting differences in AFO design characteristics and will be utilized to test the performance of various prototype designs.

The robotic surrogate presented herein demonstrated its ability to generate reliable data that replicates the types of loading experienced by the tibia during gait. (1) The robotic surrogate testing generates a low variance and high consistency of tibial strain signals, cycle to cycle, within a trial (Figure 6). (2) The strains obtained at partial anatomical load did not consistently scale to reach in vivo strain ranges, which warrants the further development of the robotic surrogate system. To address this, we will redesign the musculoskeletal apparatus to accommodate the application of full-scale loads and, if necessary, replicate anatomical force application for the entirety of the gait cycle. (3) The response of different PDE stiffnesses on strain measured by the robotic surrogate is consistent with in vivo findings and helps to explain published data on human biomechanics. These findings are also consistent with the clinical experience at the Center for the Intrepid, providing further evidence in the utility of the robotic surrogate.

Overall, these results demonstrate the potential of this robotic surrogate to be used for orthotic development in lieu of cadaveric testing and prior to human subject testing. While this work focuses on tibial stress fractures, a modified surrogate could potentially be used in the development of lower-limb orthoses for other pathologies involving bone or ligament loading (e.g., ankle fractures, ankle sprains, osteoarthritis, metatarsal stress fractures). This could be accomplished through the addition of sensors on joints or other bones and/or additional actuated degrees of freedom (e.g., ankle inversion/eversion). Moreover, a non-actuated Achilles tendon version of the robotic surrogate could be utilized to test if the axial unloading of the foot via the proximal cuff of an AFO potentially contributes to tibial strain changes. Our future work will specifically utilize the surrogate to optimize a novel AFO to reduce tibial strain and expedite a return to activity for athletes and Service members.

## Figures and Tables

**Figure 1 bioengineering-11-00490-f001:**
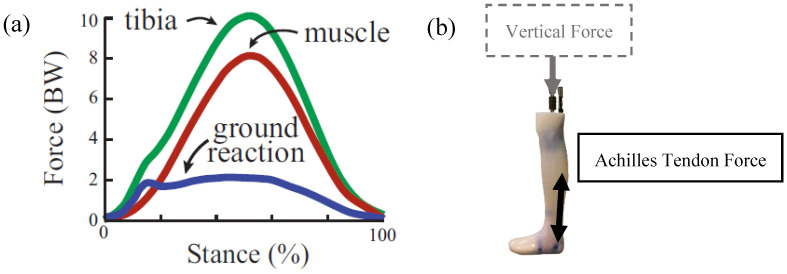
(**a**) Forces during running from Figure 1a of Matijevich et al. 2019, and (**b**) the dual loading concept for the robotic surrogate involving vertical force applied distally through the proximal femur and the Achilles tendon force applied at the posterosuperior aspect of the calcaneus.

**Figure 2 bioengineering-11-00490-f002:**
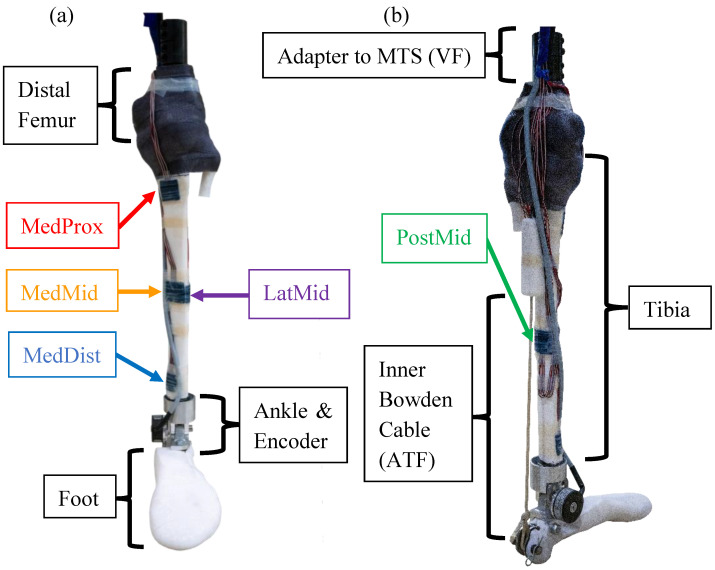
(**a**) Anterior and (**b**) medial/posterior views of the musculoskeletal apparatus prior to being encased in silicone.

**Figure 3 bioengineering-11-00490-f003:**
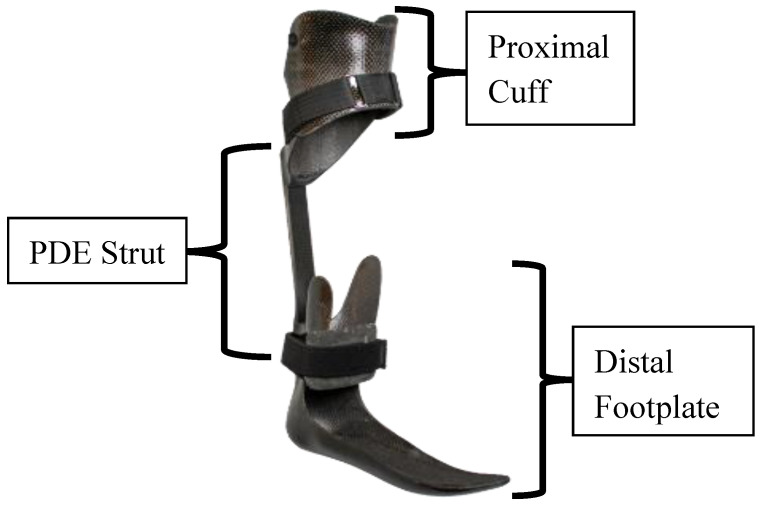
The Intrepid Dynamic Exoskeletal Orthosis (IDEO).

**Figure 4 bioengineering-11-00490-f004:**
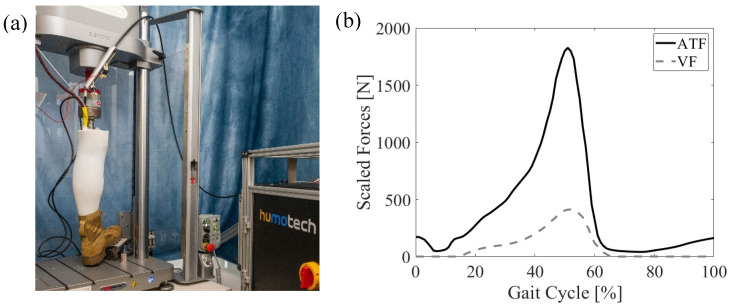
(**a**) The experimental setup of the boot-only condition in which the robotic surrogate is loaded in the MTS with the off-board Bowden cable actuator and (**b**) the applied Achilles tendon and vertical loads at 50% anatomical scale for walking.

**Figure 5 bioengineering-11-00490-f005:**
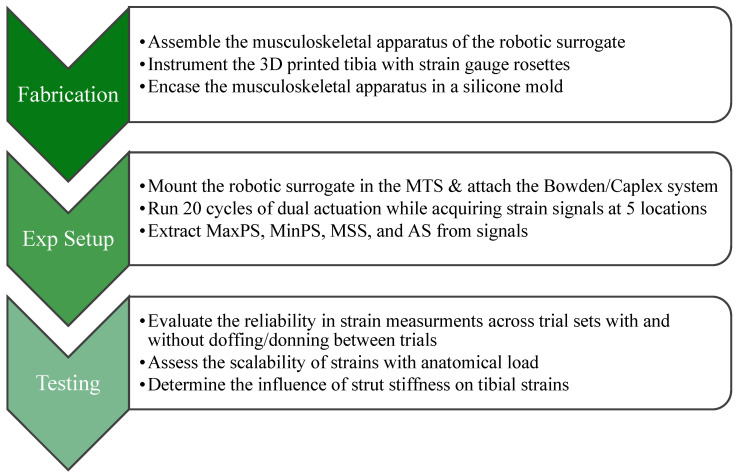
A flowchart illustrating the study methodology.

**Figure 6 bioengineering-11-00490-f006:**
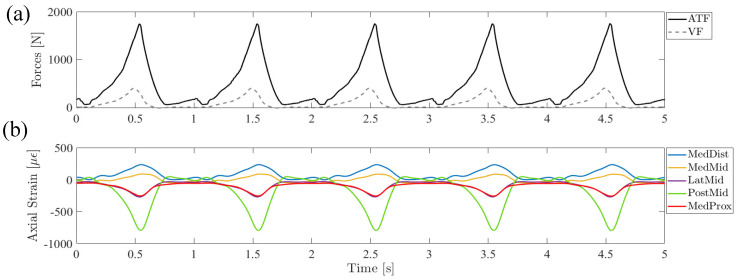
(**a**) Applied Achilles tendon and vertical loads at 50% anatomical scale and the resulting (**b**) axial strain signal time series at the five specified tibia locations from a sample boot-only trial.

**Figure 7 bioengineering-11-00490-f007:**
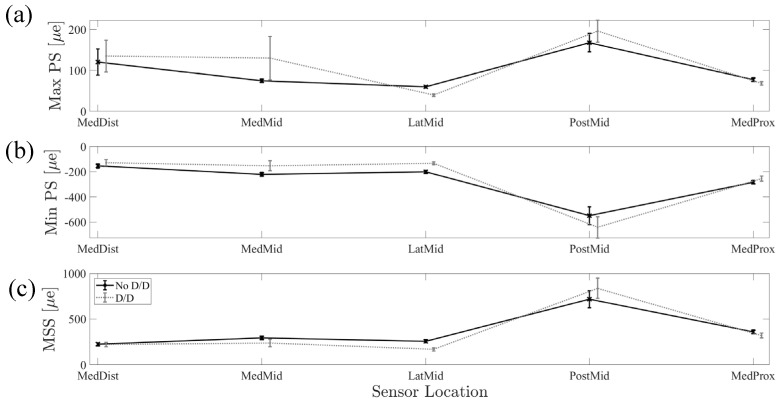
Comparison of no doff/don trials and doff/don trials across sensor locations for (**a**) Max PS, (**b**) Min PS, and (**c**) MSS, which indicate the repeatability of the experimental setup despite doffing and donning procedures.

**Figure 8 bioengineering-11-00490-f008:**
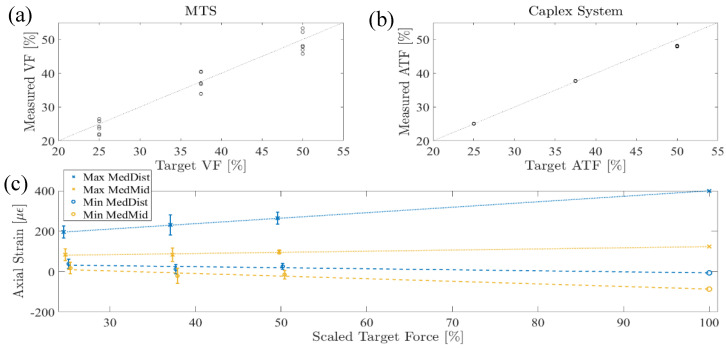
(**a**) MTS applied peak VF validation, (**b**) Caplex system applied peak ATF validation, and (**c**) measured Max and Min MedDist and MedMid strains and the corresponding strain values extrapolated at 100% of anatomical load. Error bars indicate the standard deviation for each condition.

**Figure 9 bioengineering-11-00490-f009:**
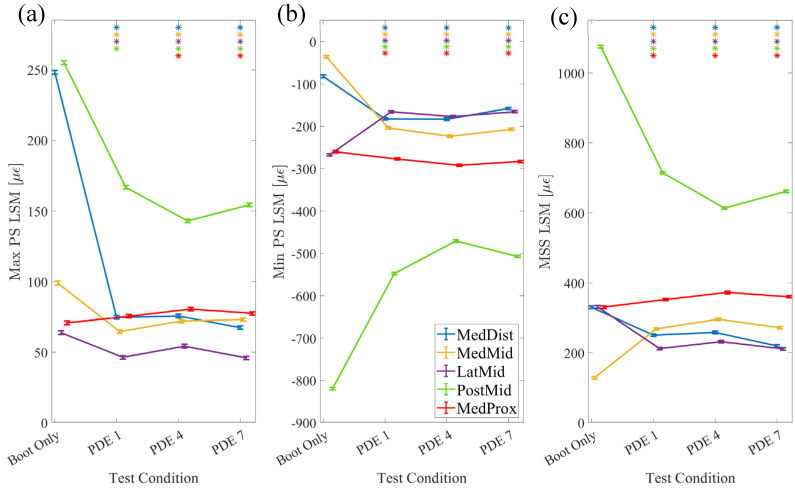
The (**a**) Max PS, (**b**) Min PS, and (**c**) MSS two-way interaction plots for statistical models depicting the differences in tibial strain across strain location and exoskeletal condition. Asterisks indicate a significant difference in strain between PDE conditions compared to boot only, within the sensor location. Error bars indicate the standard deviation for each condition.

**Table 1 bioengineering-11-00490-t001:** Extrapolated axial strain extrema for all five sensor locations.

	MedDist (µε)	MedMid (µε)	LatMid (µε)	PostMid (µε)	MedProx (µε)
Maximum	399.7	123.4	−58.4	51.8	−128.7
Minimum	−6.4	−86.7	−499.0	−1514.8	−548.3

## Data Availability

The data underlying this article will be shared upon reasonable request to the corresponding author.

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
