# Peer review of "Advancing Exoskeleton Development: Validation of a Robotic Surrogate to Measure Tibial Strain"

_bioengineering, 2024, doi:10.3390/bioengineering11050490_

Round 1

Reviewer 1 Report

Comments and Suggestions for Authors

Your research offers significant contributions to bioengineering and the advancement of orthotic devices. To further enhance the paper and prepare it for publication, the following recommendations are provided:

Illustrate the potential implications of the research on future orthotic designs and rehabilitation protocols.

Reflect on how this research could be extended or applied in different contexts or populations.

Contemplate including a section on potential commercial or clinical partnerships for real-world application.

To enhance the paper's relevance and authority, please update the references to include the most recent literature from 2021 to 2024.

The additional visual representations that could benefit the paper may include:

Strain Measurement Graphs: Detailed graphs showing the strain measurements across different points on the surrogate leg could help visualize the variance and consistency of the readings.

Surrogate Leg Design Schematics: Diagrams or CAD images that illustrate the design and the specific parts of the robotic surrogate leg would provide a clearer understanding of the experimental setup.

Load Application Diagrams: Visuals that represent how the loads are applied and distributed across the tibia during testing could be informative.

Stiffness Evaluation Charts: Charts or graphs illustrating how different stiffness settings of the AFO affect tibial strain would be useful for direct comparison.

Statistical Analysis Outputs: Any statistical analysis, such as box plots or confidence intervals, that provide a visual representation of data reliability and variability.

Comparative In Vivo Data: If available, side-by-side graphical comparison of the surrogate data with in vivo results to highlight similarities or differences.

Flow Diagram of Methodology: A flowchart showing the steps taken in the experimental process from setup to conclusion for a quick overview of the method.

Photographs: High-quality photographs of the experimental setup during different phases of testing can offer a more tangible sense of the physical aspects of the research.

Incorporating these types of visuals can make the results more accessible and easier to grasp for readers, particularly those who may be more visually oriented.

The conclusions are in line with the presented data, but acknowledging the scaling discrepancies and providing a roadmap for future research in this area would be advantageous.

 Your research provides valuable insights into the field of bioengineering and the development of orthotic devices, presenting a clear pathway for impactful future work.

Illustrate the potential implications of the research on future orthotic designs and rehabilitation protocols.

Reflect on how this research could be extended or applied in different contexts or populations.

Contemplate including a section on potential commercial or clinical partnerships for real-world application.

The additional visual representations that could benefit the paper may include:

Strain Measurement Graphs: Detailed graphs showing the strain measurements across different points on the surrogate leg could help visualize the variance and consistency of the readings.

Surrogate Leg Design Schematics: Diagrams or CAD images that illustrate the design and the specific parts of the robotic surrogate leg would provide a clearer understanding of the experimental setup.

Load Application Diagrams: Visuals that represent how the loads are applied and distributed across the tibia during testing could be informative.

Stiffness Evaluation Charts: Charts or graphs illustrating how different stiffness settings of the AFO affect tibial strain would be useful for direct comparison.

Statistical Analysis Outputs: Any statistical analysis, such as box plots or confidence intervals, that provide a visual representation of data reliability and variability.

Comparative In Vivo Data: If available, side-by-side graphical comparison of the surrogate data with in vivo results to highlight similarities or differences.

Flow Diagram of Methodology: A flowchart showing the steps taken in the experimental process from setup to conclusion for a quick overview of the method.

Photographs: High-quality photographs of the experimental setup during different phases of testing can offer a more tangible sense of the physical aspects of the research.

Incorporating these types of visuals can make the results more accessible and easier to grasp for readers, particularly those who may be more visually oriented.

The conclusions are in line with the presented data, but acknowledging the scaling discrepancies and providing a roadmap for future research in this area would be advantageous.

Comments on the Quality of English Language

The quality of English language in the manuscript is generally good; however, there are occasional grammatical errors and awkward phrasings that could be refined to improve readability and ensure precise communication of the research findings.

Reviewer 2 Report

Comments and Suggestions for Authors

The submitted manuscript describes the development of an actuated robotic surrogate leg to explore how tibial strain changes with different ankle-foot-orthosis conditions. This is a topic of interest to researchers in the relevant field, but significant improvements are needed before the paper can be accepted for publication. My detailed opinions are as follows:

1. The choice and fidelity of keywords is deficient. The first three keywords should generic and the remaining three paper specific.

2. Correct the image error in Figure 1 (b). In addition, many images in the article lack the necessary textual explanations. Please provide complete information.

3. It is noted that your manuscript requires careful editing by someone with expertise in technical English editing, paying particular attention to English grammar and sentence structure. This will ensure that the goals and results of the study are clear to the reader.

4. Currently, how is the tibial strain measured under replicated anatomical loads of BSI orthotics? Lack of relevant research status analysis in the introduction section.

5. Lack of analysis and testing on the rationality and reliability of the robotic surrogate design before measuring the surface strain of the tibia.

6. Please clarify the accuracy of the VF data for your mechanical testing system.

7.Figures require a higher level of accuracy and more detailed information. Graphs need to be improved by adding a legend (a), positioned just above the top left corner of the figure panels in the results section.

8. The analysis methods and processes in the results section need to be more detailed.

Comments on the Quality of English Language

Moderate editing of English language required.

Round 2

Reviewer 1 Report

Comments and Suggestions for Authors

Thank you for submitting the revised version of your manuscript. I have reviewed the changes made in response to the previous comments, and I am pleased to see that the suggested improvements have been thoughtfully addressed. The manuscript has been significantly enhanced and now meets the publication standards.

I recommend the paper for publication in its current form. Congratulations on the fine work.

Comments on the Quality of English Language

The English language quality of this review paper is generally good, with clear articulation of complex ideas. However, there are occasional grammatical errors and awkward phrasings that could be revised to enhance readability and maintain a professional tone.